# Trustworthy High-Performance Multiplayer Games with Trust-but-Verify Protocol Sensor Validation

**DOI:** 10.3390/s24144737

**Published:** 2024-07-21

**Authors:** Alexander Joens, Ananth A. Jillepalli, Frederick T. Sheldon

**Affiliations:** 1School of Electrical Engineering and Computer Science, Washington State University, Pullman, WA 99163, USA; alexander.joens@wsu.edu (A.J.); ananth.jillepalli@wsu.edu (A.A.J.); 2Department of Computer Science, University of Idaho, Moscow, ID 83844, USA

**Keywords:** cheating sensors in games, client/server, referee, security, gaming, behavior-based trust sensors

## Abstract

Multiplayer online video games are a multibillion-dollar industry, to which widespread cheating presents a significant threat. Game designers compromise on game security to meet demanding performance targets, but reduced security increases the risk of potential malicious exploitation. To mitigate this risk, game developers implement alternative security sensors. The alternative sensors themselves become a liability due to their intrusive and taxing nature. Online multiplayer games with real-time gameplay are known to be difficult to secure due to the cascading exponential nature of many-many relationships among the components involved. Behavior-based security sensor schemes, or referees (a trusted third party), could be a potential solution but require frameworks to obtain the game state information they need. We describe our Trust-Verify Game Protocol (TVGP), which is a sensor protocol intended for low-trust environments and designed to provide game state information to help support behavior-based cheat-sensing detection schemes. We argue TVGP is an effective solution for applying an independent trusted referee capability to trust-lacking subdomains and demands high-performance requirements. Our experimental results validate high efficiency and performance standards for TVGP. We identify and discuss the operational domain assumptions of the TVGP validation testing presented here.

## 1. Introduction

Online multiplayer video games have grown in popularity; so has the desire to cheat at them. Cheaters have a variety of motivations, including to gain a competitive edge or for financial gain by selling desirable accounts or virtual goods for real-world currency. Cheating can be highly profitable but can make the game less fun for honest players and could cause financial damage for game developers, leading to an escalatory cycle of patching, exploit–discovery between cheaters and game developers [1].

In a multiplayer video game, the key challenge is maintaining a common game state. The current state must be synchronized among all players and allow all players to input their own commands. Conflicting actions, such as two players attempting to occupy the same space, must be resolved. Modern multiplayer games almost exclusively use a client-server model, with some using a “player-hosting” model that selects one player to also be the server. In both cases, the server acts as arbitrator, deciding the current game state and keeping clients informed. Many games update at a fixed interval, with tick rate referring to the number of updates per second, and process inputs in the order in which they are received. Less interactive games can get away with lower tick rates, but highly responsive games can have tick rates of 64 or 128, updating every 15.6 ms or 7.8 ms, respectively [2].

Gameplay commonly has a real-time requirement in online multiplayer video games. The real-time feature of multiplayer games produces a scenario that involves many-many relationships among the components involved. These relationships are exponential in nature, where one action by a component creates a cascading effect impacting a multitude of other aspects. As such, games with real-time requirements are particularly challenging to secure for a cheat-sensor system [3]. One of the goals of a cheat-sensor system is to identify and, if possible, prevent tampering with the game resources, including, but not limited to, gameplay, network transmissions, player messages, and latency. The real-time multiplayer gameplay domain requires high responsiveness and low latency, in an environment where the client cannot be trusted. Various approaches exist to address the real-time requirement paradigm [3].

The current ‘holy grail’ for gameplay security is to give clients no say in deciding the game state sensing. Clients can issue commands, such as key presses, and the server determines the outcome. Such a methodology is used by the source engine, which powers the popular game Counter Strike: Global Offensive (CS: GO). Movement is smoothed out using client-side input prediction for the player and entity interpolation for everything else. When calculating the results of an action, such as firing a shot, the server uses both packet latency and entity interpolation to “unwind” the server to the time the player claims to have acted, thus simulating what was on their screen at the time when deciding the results [4].

In general, the more responsive a game is, the harder it is to secure, and the most responsive games cannot operate on slower, more secure protocols. Among other limitations, the physical distance between players cannot be traversed faster than the speed of light, putting a hard limit on how responsive a game can be. To overcome this, we must allow the game state to be slightly desynchronized between clients [5]. However, for the same reasons we made these design compromises in the first place, we are limited in our ability to validate client behavior.

Therefore, for this sensing model to work, we must be able to trust the client. A common solution is to require players to run separate cheat-sensor software alongside their game. However, alluding to the arms race, modern cheat-sensing software has evolved to be increasingly intrusive. Games like Valorant use kernel-level cheat-sensing tools that have hardware-level access to system memory, effectively circumventing any user-mode protections.

### 1.1. Research Problem

Game developers are confident in their security practices and in their ability to respond to threats [6]. Kernel-level cheat-sensor systems present an appealing attack vector and thus a major security vulnerability [6]. With active player counts in the millions, these games and their anti-cheat systems have large install bases, making them increasingly attractive targets. We believe that it is a matter of time before a vulnerability in a kernel-level cheat-sensor tool is exploited and people’s digital information is exposed on a large scale.

Several games have only a few layers of security—once a cheat-sensor measure is circumvented, there is little to stop a potential hacker. In-game reporting tools are a final layer of defense but are often slow to respond, giving cheaters plenty of time to ruin the experiences of other players. Also, in a player-hosting model, server-side sensing measures are ineffective, as server software can be compromised in the same ways as client software.

### 1.2. Core Challenge

The core challenge is to simultaneously provide integrity and high performance. A key observation is that these can be accomplished separately. A potential solution is behavior-based cheat sensing, which monitors player and server behavior to detect abnormalities. As an analogy: instead of removing every potentially flammable item from a room, we can install a smoke detector, which can identify smoke generated regardless of the material/nature of a fire. Similarly, behavior-based cheat sensors identify anomalous behaviors regardless of the cause and/or actor involved [7].

A behavior-based cheat-sensing method of cheat sensing reduces the attack surface dramatically, as instead of having to identify and seal potentially thousands of exploitable vulnerabilities in a game’s software, we need only to monitor the result. This shifts the focus from eliminating cheating to reducing its effectiveness, as the former may be intractable. Ultimately, such an approach more directly addresses the goal of protecting the gameplay experience.

Behavior-based cheat sensing can be used in any client-server setting but is especially useful in scenarios where the server cannot be trusted. Presently, such an ‘untrusted server’ subdomain has no effective solution beyond making the client and server software harder to compromise.

Building behavior-based cheat-sensor systems is not a trivial task and is a major challenge in and of itself. They are not currently used for online games, and it is likely that machine learning technology must advance further to make it possible. That said, any such sensor system can only operate effectively if it is given reliable and accurate information, a problem we can consider today.

### 1.3. Proposed Solutions

In this paper, we detail a new Trust-Verify Game Protocol (TVGP) that will provide game state data as a key component in future behavior-based cheat-sensor systems. We outline how such a model might be implemented, as well as the domains for which it is best suited. We identify and detail specific security vulnerabilities present in current online multiplayer games and how TVGP can be used to protect against them.

TVGP is game agnostic. Our sensor model will be applicable to any game. A game is any set of circumstances that has a result dependent on the actions of two or more players. A player is a strategic decision maker within the context of a game. A strategy is a complete plan of actions that a player will take within the game.

### 1.4. Outline of the Paper

The remainder of this paper is organized as follows. Section 2 summarizes the existing work in the field of gameplay security systems that are behavior-based. Section 3 details the TVGP protocol, including the protocol’s initiation, setup, connections, and transactions. Section 4 presents a security analysis of TVGP, including an analysis of TVGP in temporally both honest/dishonest environments. Section 5 discusses our experiments for testing sensor detection rate and performance using proof-of-solution implementation of the TVGP system. Section 6 lists the operational domain assumptions of TVGP, including the limitations and potential future work for the TVGP protocol. Finally, Section 7 offers our conclusions, followed by acknowledgements and references.

## 2. Related Work

There is little publicly available research in the field of behavior-based cheat sensors. Modern anti-cheat systems are developed by private companies, which have a vested interest in keeping the inner workings of their security systems private. Here, we list some tangentially related works and the differences between such works and TVGP.

Webb et al. [7] described a conceptually similar protocol, Referee Anti Cheat Scheme (RACS). They focused on a peer-to-peer (P2P) architecture, in which players can message each other directly, and imposed a central authority that effectively acts like a server. The referee receives messages from the players, maintains its own game state, and corrects any detected inconsistencies.

TVGP and RACS share the same high-level approach, as well as a few implementation details, but are substantially different in both conception and execution. RACS is as much a method for synchronizing game states as it is about providing security; TVGP, paired with a client-server model, can optimize for both tasks separately. RACS can potentially be more responsive than a traditional client-server model, as players can message each other directly but only if messages are received by players and the referee during the same time window. If not, the referee must detect then send a correction, which is slower than the traditional model. This effect worsens with higher tick rates, making it less efficient than a client-server model in some high-performance scenarios.

Cronin et al. [8] described a new network architecture, which replaces a single server with a private P2P network, using a trailing state synchronization protocol to keep game states consistent at the edge nodes. Players connect to the closest peer, or ‘Mirrored Server’, potentially offering better latency than a traditional client-server model. Their paper did not consider security. It is only peripherally related to TVGP, only sharing a handful of design criteria.

Several of the literature items have documented networking [9,10], scalability [11,12], and load combinatorial issues [13,14] in multiplayer or Massively Multiplayer Online (MMO) gaming environments. These works also tackled cheating issues arising from a technological perspective such as packet sniffing [15,16], code injection [17,18], ping fuzzing [19,20], aim-bots [21,22], trigger-bots [23,24], and lag-switching [25,26]. These works also dealt with issues in support mechanisms offered to gamers [27]. Our proposed work deals with behavior-based cheat sensing and can work symbiotically alongside some or many of the works presented in this section.

In the existing ecosystem, there is a lack of algorithms or models that aim to address cheating issues arising from a behavioral perspective such as syndication, spear phishing, group targeting, and so on. Current solutions for behavior-based cheating issues are manual in nature, where a group of moderators are given the responsibility of monitoring game environments to identify and disqualify any actors that exhibit misbehavior. TVGP aims to automate the process of detecting and handling behavior-based cheating or malicious activities.

## 3. Trust-Verify Game Protocol

This section details the Trust-Verify Game Protocol (TVGP) and is structured as follows. Following an introduction, we cover preparing the model, initiating a game, and the core gameplay loop, split into two sections. Figure 1 presents a pictorial representation of the TVGP system at a high level.

TVGP operates on a ‘trust but verify’ principle. TVGP’s core goal is to provide security with zero penalty to responsiveness, for both the client and the server. All additional processing and messaging are offloaded to separate threads, which will not block the normal execution of the game.

### 3.1. Preparation

TVGP requires a trusted authentication server TA, which can be used to authenticate and securely communicate with players. In the context of online games, login servers are often used. The players $P=\{P_{a},P_{b},\ldots ,\;P_{k}\}$ all use the TA authentication server to establish their identities. TA sends the list of authenticated players to a trusted matchmaking server TM. Players would be able to create game lobbies and configure their matchmaking specification via the server TM.

A match (game session), M, is created when a satisfactory number of players are identified, for any given set of matchmaking specifications. A referee R is a process on a trusted host, separate from the server S and other entities. The server S manages the game state, receiving player commands and issuing updates. In a player-hosting model, one of the players may be selected as the server.

### 3.2. Initiation

When a game is initiated, TM will assign a referee R to the match. The referee R generates one set of keys for every player participating in a match. The set of keys contain hash function {H}, public encryption keys $\left\{E\right\}$, and corresponding decryption keys $\left\{D\right\}$. The set of keys are assigned and securely transmitted by *R*. The server will receive an encryption key $E_{S}$.

At the end of the key distribution process, each player will have their public key and hash function: $P_{i}∶=\left(E_{i},H_{i}\right)\;\forall P_{i}\in \left\{P\right\}$. The public key $E_{i}\;$is issued by the referee *R* to a specific player *P_i_*. The public keys of players are not made available to all other players in a match. The referee R has these as well as the corresponding private key $D_{i}$ for message decryption: $R∶=\left(E_{i},H_{i},D_{i}\right)\;\forall i\in \left|\left\{P\right\}\right|$. The server S has its own public key: $S∶=(E_{S})$, which the ref can decrypt: $R∶=(E_{S},D_{S})$.

### 3.3. Gameplay

In a client-server model, we can break the main interaction loop into two phases: the player issuing a command and the server sending the result. Below, we explain the process for a single player, which is repeated for every player.

### 3.4. Client to Server

When a player $P_{a}$ has content c that it is ready to send to the server, it uses the player identifier a and a counter x to create a message identifier $i=\left(a,x\right)$. The message $M_{S}=\left(i,\;c\right)$ is sent immediately; then, its contents are copied to a separate thread $t_{R}$. The $t_{R}$ gets a current timestamp $t_{c}$, generates a salt s, and hashes the message $h_{P}=H\left(H_{a},\left(c,\;s\right)\right)$. These are combined and encrypted to form a message $M_{R}=E\left(E_{a},\left(t_{c},i,s,h_{P}\right)\right)$ that is sent to the referee R. The items *H_a_* and *E_a_* are the hash function and public encryption key for a given player, $P_{a}$. They are placed in their respective packets to better allow the server’s association of the packets to a player’s identity.

When a server receives a message $M_{S}=(i,c)$ from a player, the server copies it to a separate thread $t_{R}$. The $t_{R}$ simply generates a timestamp $t_{s}$, encrypts the message$\;M_{R}=E\left(E_{S},\left(t_{s},i,c\right)\right)$, and sends it to the referee R.

The referee will wait to receive each of these messages, decrypting upon arrival, using the identifiers i to match server messages $M_{S}=\left(t_{s},i,c\right)$ to client messages $M_{C}=\left(t_{c},i,s,h_{P}\right)$. The referee logs the timestamps $\left(t_{c},t_{s}\right)$, which indicate when the message was sent and when it was received. It uses the given salt s to hash the content provided by the server $h_{R}=H\left(H_{a},\left(c,s\right)\right)$ and compares the hash to the given one $\left[h_{P}==h_{R}\right]$. If they match, we can be sure that the player sent c and it was correctly received. The referee can now log c and $\left[h_{P}==h_{R}\right]$ and use them for analysis. An illustration of the client-to-server communication process is given in Figure A1 (see Appendix A).

### 3.5. Server to Client

The server has content c intended for the player Pa and a unique identifier i. The server composes a message $M_{S}=\left(i,c\right)$ and sends it to the player as normal. The message is also copied to a separate thread, which generates a timestamp, encrypts the message, and sends it to the referee R. Upon receiving a message $M_{S}$ from the server, the client copies it to a separate thread $t_{R}$ and processes the message like normal. The $t_{R}$ follows a similar process to phase one, except it is hashing the content c that it received from the server and it will not salt it.

The referee will follow the same procedure as before, matching the server’s stated message contents to the client’s hash, verifying that the server sent what it claimed to and the player received it. This protocol provides a referee with the following information: (i) every player-issued command, (ii) every server-issued update, (iii) when these messages were sent and received, and (iv) verification that all messages were correctly received. The above information is sufficient to accurately monitor the server and ensure that it is not behaving inappropriately. If any message fails verification, this indicates tampering.

## 4. Security Analysis of Trust-Verify Game Protocol

The simplicity of TVGP belies its robustness. We provide a general analysis, which describes high-level properties and benefits; a detailed breakdown of how standard methods of tampering will be detected; and a consideration of temporal attacks.

### 4.1. General Analysis

The referee is designed to operate at a slight delay, which means both players and the server can send their messages after the core game messages have been sent. Though the protocol generates more system load, it should not introduce any latency, which is to say that TVGP prioritizes latency optimization at the expense of higher computational minimum requirements of a system.

When compared to peer-to-peer systems, TVGP has the distinct advantage of not exposing the IP addresses of other users, as these allow vindictive users to identify and locate other players in real life. Real-life retribution for perceived slights in online games is a known issue [28,29]. One of TVGP advantages, when compared with peer-to-peer systems, is protecting users’ IP addresses. Thereby, we minimize the risk of exposing the location of the system users, which in turn minimizes the risk of potential retribution.

The protocol gracefully degrades with nonparticipation. If a single player sends false messages or none, the referee will detect this and action can be taken. In a competitive environment, this might result in the player being removed, but in a casual one, they may simply be tagged with a visual indicator that other players can see. If the server does not participate, players will also be notified; in a player-hosting model, this might lead to a new host being automatically selected and all players automatically transferring over.

As detailed in the Testing section (Section 5), TVGP reliably detects three forms of tampering. They are ping-lag exploit, fabrication exploit, and sybil exploit (both moderated and unmoderated versions). Section 5.1 contains a detailed discussion of how TVGP is able to differentiate among these three types of exploits. If even a single player is concerned, they will always be informed of nonparticipation and/or cheating. TVGP does break down if all players and the server refuse to participate. However, this can be argued as an irrelevant scenario: cheat sensing is performed largely for the benefit of the player, and if none of the players care, then it does not matter if TVGP fails. This is an issue that can easily be designed around, such as not granting rewards for games played on unmonitored servers.

To evaluate the system, we shall consider two scenarios, each from the perspective of an honest player and a cheater. Both cases establish that with TVGP, malicious behavior can always be detected.

### 4.2. Honest Server

Let us consider a dishonest player who has already compromised the client software module on their side and is attempting to issue a malicious command. They must send a command to the server and a command hash to the referee. They can mask their activity with an honest command. If they send the malicious command and the malicious hash, then the referee will verify that the command was properly sent and use it in its sensor model. From here, the referee can identify the cheating and take appropriate action.

If they send a mismatched command and hash, the command will fail to be verified, which would never happen under normal circumstances. It indicates that either the server or the player is being dishonest, and further analysis can identify. If they send an honest command and the honest hash, we have the highest probability of detecting a cheater. If the server is trusted, we can automatically assume any inconsistent behavior is because of the player cheating, so the player can be immediately sanctioned. If the game design allows for it, the malicious activity can even be undone.

### 4.3. Dishonest Server

It is possible for both a server and client to be compromised, especially in the player-hosted model. Likely, the player will attempt to use this to their benefit. As before, we assume the player has a malicious command as well as an honest command to choose from, but now the server can lie about which message it received. Let us consider the possible outcomes. If they send the malicious command and hash, the referee will verify that the message was properly sent. This will incriminate the player. If the hash and message are mismatched, the referee will detect this and can perform further analysis to narrow down the misbehaving agent or processes. If they send the honest command and hash while the server secretly processes the malicious command, the referee will verify the message but detect that the server responded incorrectly.

A dishonest server can also sabotage and falsely incriminate other players by substituting their honest command with a malicious one. Anomalous behavior is not always the fault of the player. If the server sends a malicious message, then the referee will fail to verify the command, indicating that something is wrong. If the server sends an honest message but acts on the malicious one, anomalous server behavior will be detectable. Because these attacks exist, while we can detect misbehavior, we cannot always identify the culprit when the server misbehaves or a command fails verification. TVGP can notify all players that anomalous activity has been detected, informing an honest player that their experience may be affected and a cheating player that they have been found out. However, in this environment, we cannot automatically make any decisions.

### 4.4. Temporal Attacks

A specific attack vector used by malicious player hosts is time. A cheater can use a “lag switch” to artificially increase the latency of their internet connection, slowing down the connections of all other players and giving them an edge. This is the reason for recording timestamps. However, the system is not perfect, as timestamps for inbound and outbound messages can be forged.

The referee will need to monitor a player for several matches to determine if their latency is substantially different when they are hosting. It is unwise to punish the player, as the latency may simply be due to playing on a low-bandwidth connection, but the degraded experience can be detected and that player can be selected as host less often in the future.

## 5. Testing

Gaming companies are concerned about keeping their security systems obfuscated from public research communities. As such, it is a challenging task to test experimental security frameworks in production and/or pre-production environments of private games.

After a long search and negotiation process, we were able to obtain testing access on one of the web browser production environments for the game ‘territorial.io’, hereby referred to as the sponsor. We built a proprietary TVGP proof of solution tailored to the security mechanism of the sponsor. We then conducted a series of experiments to identify the effectiveness of the TVGP system.

### 5.1. True Positive Experiment Setup

We randomly selected a group of 300 anonymous users online out of the web browser player base of ‘territorial.io’. Informed consent was obtained from the anonymous users for their participation in the experiments. Let us label this user group as ‘TestMal’. First, we asked ‘TestMal’ users to attempt a series of prescribed attacks continuously over a period of 60 min.

The prescribed attacks were behavioral in nature and three in number. At this time, we are unable to test TVGP with a wider variety of attack vectors due to the limitations imposed by this study’s sponsor. With additional funding or a willing volunteer company as a sponsor, a future study could expand the scope of tested attack vectors and users involved.

The three behavioral exploits that we were able to test were as follows: ping-lag exploits, fabrication of territories exploit, and sybil exploit. In the ping-lag exploit for this game, the player could run a script that repeatedly disconnected and connected the player computer’s network adapter. Such a behavior caused the player’s territory to not shrink when under assault from other players. In the fabrication of territories exploit, the player used turbo-triggering of the assault key. Such a behavior caused the player to fabricate the territory of another player as their own without engaging in combat. Finally, in the sybil exploit, a group of users that knew each other in real life tricked the server into placing them all in the same match such that they were at least 51% of the players in a match. Such a behavior allowed the exploiting players to gang-up against the other players, one at a time, which gave the exploiters a major advantage. For the first true positive iteration, we used the default security system of the sponsor. The first iteration is hereby referred to as the control true positive experiment. After a break of 60 min, we asked ‘TestMal’ users to play the game and repeat the same prescribed attacks continuously for a period of 60 min. However, for the second true positive iteration, we deployed the TVGP proof of solution on the sponsor’s default security system. The second true positive iteration is hereby referred to as the test true positive experiment.

After concluding both the control and test experiments, we surveyed the ‘TestMal’ users to record attack frequency. From the post-experimental survey, we know that every ‘TestMal’ user attempted at least 27 exploits each for both the control and test experiments. For benchmarking purposes, we assumed the minimal exploit number as the baseline. Therefore, we ended up with 8100 exploits from 300 users across 60 min in both control and test experiments.

The post-experimental survey also informed us that, out of the 8100 attacks, ping- lag exploits were most frequent, at 4926 attacks, followed by fabrication exploits, at 1956 attacks. The lowest frequency of attempted attacks was for sybil exploits, at 1218 attacks. We speculate that moderator monitoring was the reason for a low frequency of sybil attacks since the presence of a moderator discourages syndicate behaviors. An unmoderated session did show a higher incidence of sybil exploits. The control detection was significantly lower than the TVGP detection of sybil exploits in an unmoderated session.

### 5.2. True Positive Experiment Results

The experiment results indicate that the test true positive detection rate was better than the control true positive detection rate for two of the three prescribed attacks. TVGP was able to better detect ping-lag exploits and sybil exploits when compared to the sponsor’s default security system. However, TVGP’s detection rate was lower than the default security system’s detection rate for fabrication exploits.

Such an underperformance for fabrication attacks was expected due to the referee-based behavioral detection scheme of TVGP. Fabrication exploits are primarily based on packet sniffing, manipulation, and replaying. TVGP is not designed to detect packet manipulations. Table 1 presents true positive detection rates for control (default security system) and test (TVGP overhead implementation) experiments.

### 5.3. False Positive Experiment Setup

We randomly selected a group of 300 anonymous users online out of the web browser player base of ‘territorial.io’. Let us label this user group as ‘TestCln’. To keep the ‘TestCln’ group separate from ‘TestMal’ group, we requested the users of ‘TestMal’ group not to join the ‘TestCln’ group. However, since the users were completely anonymous, we had no way of verifying if any users from the ‘TesMal’ group rejoined the ‘TestCal’ group.

First, we asked ‘TestCln’ users to play the game without any malicious behavior for 60 min. We emphasized that the ‘TestCln’ group should intentionally avoid any kind of malicious or abusive behavior during their gameplay. For the first false positive iteration, we used the default security system of the sponsor. The first false positive iteration is hereby referred to as the control false positive experiment. After a break of 60 min, we asked ‘TestCln’ users to play the game again for a period of 60 min by intentionally avoiding malicious or abusive behavior. For the second false positive iteration, we deployed the TVGP proof of solution on the sponsor’s default security system. The second false positive iteration is hereby referred to as the test false positive experiment.

### 5.4. False Positive Experiment Results

The experiment results indicate that the false positive sensing rate was similar across both control and test false positive experiments. Control was slightly better than test, at 324 vs. 351 false positives detected. We speculate that these false positive instances may have occurred due to either a) some users of ‘TestCln’ did not follow the directions they were given and performed malicious activities anyway and/or b) the systems correctly recognized an occurrence as malicious or abusive but the user did not perform that action consciously with malicious intentions.

An example of the latter phenomenon is ‘rage-clicking’ [30]. ‘Rage-clicking’ in this case occurred when a user is frustrated with their in-game situation and tried to click their way out of a bad situation. Instances of ‘rage-clicking’ are usually detected as ping-lag exploits. Table 2 presents false positive sensing rates for control (default security system) and test (TVGP overhead implementation) experiments.

### 5.5. Performance Results

We tracked performance metrics through both the false and true positive experiments both on the server side and the client side using the sponsor’s game. Since our TVGP implementation was deployed as an overhead module rather than an in-built module, we expected TVGP implementations to take a small performance hit. Our observations detected results that matched our expectations. Table 3 and Table 4 present the network, processing, and memory performance metrics of all four of our experiments both from server- and client-side perspectives.

We can observe that the difference in performance metrics is manageable in its current state. Such a difference in performance rates was due to the TVGP implementation being unoptimized. Our goal here was to test for validity and not to produce an optimized, commercially viable product. We are confident that a commercial deployment of TVGP would be seamless in terms of performance when compared to default security implementations.

Specific optimization strategies will be dependent on the underlying game that intends to utilize TVGP. As a game agnostic protocol, TVGP is already a bare-bones, minimalistic model. For the game that we used for testing, TVGP’s additional overhead per message was 192 bytes. This overhead could be optimized further at the expense of reducing hashing function size. We used 512-bit hashing for the tests above; if we were to downgrade to 128-bit hashing, the overhead of TVGP could be reduced to 96 bytes. However, given the current computational and networking capabilities of an average multiplayer gamer, we decided to not sacrifice the security of our implementation for lower overhead.

## 6. Operational Domain Assumptions

Several unexpected behaviors arise when TVGP operates in a state that is different from our operational domain assumptions. As TVGP is designed to protect the player’s experience, these behaviors are acceptable if TVGP can accomplish its primary objective. However, if attempting to utilize TVGP for other means, such as protecting gameplay integrity, the unexpected behaviors would be considered flaws.

Some operational domain assumptions are as follows: The referee operates at a small delay, meaning it cannot stop the cheating and cannot act immediately. The impact of the cheat itself is dependent on game design and may be substantial. However, in the context of multiplayer gaming, verification occurring within seconds or fractions of a second is acceptable and a monumental improvement over existing systems that take minutes or hours to respond.

Race conditions and deliberate collisions could allow a player in a match to override valid messages. Likewise, a third-party interceptor that is in possession of a match’s public key can also override valid messages using race conditions and deliberately crafted collisions. TVGP does not work if all players and the server refuse to participate. Sometimes, players may not want strict rule enforcement, such as when playing on a server with user modification patches. TVGP ensures game security for players that desire it; if none do, it does not matter if it fails.

If a message fails verification, it may not be recoverable and, in some instances, it will not be clear whether it was the client or the server who tampered with it. Some referees may have little to no tolerance for missing messages, requiring a protocol like TCP to ensure delivery. Messages that are delayed will delay game verification. In general, there exists a discrepancy between sensing and identification. TVGP is very reliable for detecting that something is wrong but is less reliable for identifying the exact issue. This occurs often enough with untrusted servers that it is important for game designers to consider their goals in attempting to secure their game. The rest of this section considers flaws that arise when the server cannot be trusted.

When a message fails verification, it is not possible to identify who tampered with the game (client, server, or both). If unable to infer the message contents, the referee can no longer perfectly model every entity, likely reducing its effectiveness. A colluding client and server can provide the referee with fake message data and process a separate malicious one. TVGP allows this, and the referee must use behavior modeling to detect issues. However, as a server may try falsely incriminating an innocent player, unusual player behavior should not result in automatic punishment.

Temporal attacks, such as lag switches, can be made harder to detect if the server carefully forges timestamps on all sent and received messages from unfavored players. Timestamps are not inherently trustworthy, so extensive modeling may be required over several matches to identify the most sophisticated temporal attacks. Lag switches may be easily identified, but this attack vector otherwise remains open.

An interesting new attack exists if the referee is naïve. In a player-hosted model, as most unidentifiable attack types involve the server, the referee could automatically switch the host if any anomalous behavior is detected. However, if host switching has any desirable in-game effects, a cheater might intentionally send a faulty message to the referee and trigger a host switch whenever it is advantageous. In general, a referee-based system would change the nature of cheating but not eliminate it. We see two common strategies: dormant cheaters, who strike very rarely and only after extended periods of legitimate play, and subtle cheaters, whose cheats have small enough effects to avoid sensing. These both depend on the accuracy of the referee and will likely diminish over time.

## 7. Conclusions

TVGP has several desirable properties that make it well-suited for helping a behavior-based cheating-sensor system. In our tests, a TVGP implementation provided game state information to a cheat-sensor engine with minimal performance delays. As behavior-sensing technology improves, frameworks like TVGP would allow traditionally untrusted environments to have some degree of trust and/or security.

Even trusted servers benefit from the TVGP abstractions: a future multiplayer networking model might have separate game servers, which can be extremely widespread and created on demand, that connect to referees operating in data centers where the bulk computation is cheaper. This would offer unprecedented responsiveness with no compromises to the overall player experience. We hope our work helps behavior-based sensor system designers to design and implement cheat-sensing engines for multiplayer games.

## Figures and Tables

**Figure 1 sensors-24-04737-f001:**
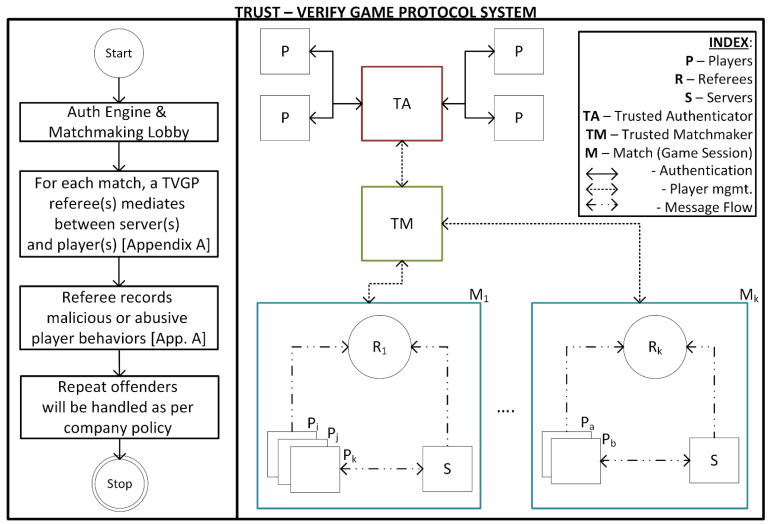
A high-level abstract of the Trust-Verify Game Protocol (TVGP) system.

**Table 1 sensors-24-04737-t001:** Control vs. test true postive sensor rates for exploits.

Exploit and Frequency	Control True Positive Sensor Rate (Approx.)	Test True Positive Sensor Rate (Approx.)
Ping lag exploit (4926 instances occurred)	63% (3103 instances detected)	79% (3891 instances detected)
Fabrication exploit (1956 instances occurred)	71% (1388 instances detected)	57% (1115 instances detected)
Moderated sybil exploit (1218 instances occurred)	77% (937 instances discovered)	86% (1048 instances discovered)
Unmoderated sybil exploit (3712 instances occurred)	28% (1066 instances discovered)	94% (3491 instances discovered)

**Table 2 sensors-24-04737-t002:** Control vs. test false postive sensor rates for exploits.

Exploit and Frequency	Control False Positive Sensor Rate (Approx.)	Test False Positive Sensor Rate (Approx.)
No exploits performed	324 false positive ping-lag exploits detected	351 false positive ping-lag exploits detected

**Table 3 sensors-24-04737-t003:** Server-side performance metrics during all four experiments.

Parameter	Control True Positive Experiment	Test True Positive Experiment	Control False Positive Experiment	Test False Positive Experiment
Network (Ping)	1–4 ms	2–6 ms	1–2 ms	1–4 ms
Processing (CPU)	2.62– 3.97 GHz	2.66–4.13 GHz	2.60–2.95 GHz	2.62–3.78 GHz
Memory (RAM)	7.41–9.26 GB	7.90–9.81 GB	7.23–8.59 GB	7.62–9.41 GB

**Table 4 sensors-24-04737-t004:** Client-side performance metrics during all four experiments.

Parameter	Control True Positive Experiment	Test True Positive Experiment	Control False Positive Experiment	Test False Positive Experiment
Network (Ping)	13–104 ms	15–122 ms	9–103 ms	15–128 ms
Processing (CPU)	1.20–2.50 GHz	1.38–2.82 GHz	1.18–2.32 GHz	1.26–2.65 GHz
Memory (RAM)	1.21–4.92 GB	1.50–5.46 GB	0.98–4.13 GB	1.12–4.80 GB

## Data Availability

The datasets presented in this article are not readily available because the game involved in this study is a proprietary commercial asset owned by Territorio LLC. Requests to access the datasets should be directed to davidtschacher@territorial.io.

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
