# Peer review of "Trustworthy High-Performance Multiplayer Games with Trust-but-Verify Protocol Sensor Validation"

_sensors, 2024, doi:10.3390/s24144737_

Round 1

Reviewer 1 Report

Comments and Suggestions for Authors

1-[Abstract, Line 10-11]: The statement "Several genres of games are thought to be difficult to secure" lacks specificity and critical context. Which genres exactly are being referred to, and why are they considered particularly challenging to secure? I don't mean to refer some citations here in the abstract section, but in the main text, still they can't be located what genres you mean. It would greatly enhance the abstract's clarity if the authors could provide concrete examples of these genres.

2-[Abstract, Line 14-15]: The claim that behavior-based security sensor schemes are "very promising" needs substantiation and more detailed explanation. What specific evidence supports this assertion? The authors should consider including brief mentions of successful implementations or theoretical advantages of these schemes to bolster their claim. I recommend changing a word that can also clarify your thoughts. This also happens in Line 85 where you mentioned 'a promising answer'. Otherwise, you need to elaborate the degree of 'promising'.

3-[Introduction, Line 31-32]: The phrase "veritable arms race" is colloquial and lacks the academic rigor expected in a scholarly article. Please rephras it using more formal language.

4-[Introduction, Line 44-46]: The statement about the "goal of a cheat sensor system" is vague and lacks the precision needed in an academic paper. Please provide a more detailed and precise definition of the system's objectives.

5-[1.1 Research Problem, Line 70-72]: The claim about kernel-level cheat sensor systems being a major security vulnerability is a significant assertion that requires more supporting evidence or citations. Your cited [6] seems to be a news report from a website. My purpose is not to seek authenticity or falsification, but at least you need to find some academic supports to help this manuscript show a better research background and research objectives.

6-[1.2 Core Challenge, Line 83-85]: The analogy of a smoke detector, while initially appealing, is oversimplified and doesn't accurately capture the complexity of behavior-based cheat sensing in online gaming environments. How does this analogy represent the multifaceted nature of detecting cheating behaviors? The authors should consider expanding this analogy or replacing it with a more comprehensive explanation that better reflects the intricacies of their proposed system. And please add some references to support this point.

7-[1.4 Outline of the paper, Line 115-121]: The outline is too brief and doesn't provide enough detail about each section's content. Please expand this section to make a clear roadmap. Consider adding a sentence or two about the key points or arguments that will be presented in each section.

8-[2. Related Work, Line 149]: Update the academic writing for this part: [9], [10], [11], [12], [13], [14], [15], [16], [17], [18], [19], [20], [21], [22], [23], [24], [25], [26], [27], 149 [28]. Using professional riting and citation methods.

Author Response

We thank the reviewer for their time and valuable feedback. We have addressed the reviewer’s feedback as follows:

Comment 1 - [Abstract, Line 10-11]: The statement "Several genres of games are thought to be difficult to secure" lacks specificity and critical context. Which genres exactly are being referred to, and why are they considered particularly challenging to secure? I don't mean to refer to some citations here in the abstract section, but in the main text, still they can't be located what genres you mean. It would greatly enhance the abstract's clarity if the authors could provide concrete examples of these genres.

Response 1 – We clarified the description by using more concise language instead of general statements. The new description used is: “Online multiplayer games with real-time gameplay are known to be difficult to secure due to the cascading exponential nature of many-many relationships among the components involved.” Citation [3] helps to support this argument. We added an elaboration to this effect in Section 1.

Comment 2 - [Abstract, Line 14-15]: The claim that behavior-based security sensor schemes are "very promising" needs substantiation and more detailed explanation. What specific evidence supports this assertion? The authors should consider including brief mentions of successful implementations or theoretical advantages of these schemes to bolster their claim. I recommend changing a word that can also clarify your thoughts. This also happens in Line 85 where you mentioned 'a promising answer'. Otherwise, you need to elaborate the degree of 'promising'.

Response 2 – Behavior-based cheat sensing is a new paradigm and there is not much conclusive evidence documented in the literature yet. Therefore, we modified the description by using more concise language instead of general statements. The new description used is: “could be a potential solution” instead of a “a very promising solution” in both the identified locations (line 14-15, line 85).

Comment 3 - [Introduction, Line 31-32]: The phrase "veritable arms race" is colloquial and lacks the academic rigor expected in a scholarly article. Please rephrase it using more formal language.

Response 3 – We modified the description by using more formal language. The new description used is: “escalatory cycle of patching, exploit-discovery.”

Comment 4 - [Introduction, Line 44-46]: The statement about the "goal of a cheat sensor system" is vague and lacks the precision needed in an academic paper. Please provide a more detailed and precise definition of the system's objectives.

Response 4 – We elaborated this statement with additional detail. The new description used is: “One of the goals of a cheat sensor system is to identify, and if possible, prevent tampering with the game resources. Including, but not limited to, gameplay, network transmissions, player messages, and latency.”

Comment 5 - [1.1 Research Problem, Line 70-72]: The claim about kernel-level cheat sensor systems being a major security vulnerability is a significant assertion that requires more supporting evidence or citations. Your cited [6] seems to be a news report from a website. My purpose is not to seek authenticity or falsification, but at least you need to find some academic support to help this manuscript show a better research background and research objectives.

Response 5 – We replaced the reference [6] with an academic paper. The new reference can be accessed at the following link: https://doi.org/10.58445/rars.657. This paper discusses the vulnerabilities in Valve’s kernel-level anti-cheat system.

Comment 6 - [1.2 Core Challenge, Line 83-85]: The analogy of a smoke detector, while initially appealing, is oversimplified and doesn't accurately capture the complexity of behavior-based cheat sensing in online gaming environments. How does this analogy represent the multifaceted nature of detecting cheating behaviors? The authors should consider expanding this analogy or replacing it with a more comprehensive explanation that better reflects the intricacies of their proposed system. And please add some references to support this point.

Response 6 – We elaborated the analogy description and added a citation to support the argument. The new description is: “As an analogy: instead of removing every potentially flammable item from a room, we can install a smoke detector, which can identify smoke generated regardless of the material/nature of a fire. Similarly, behavior-based cheat sensors identify anomalous behaviors regardless of the cause and/or actor involved [7]”.

Comment 7 - [1.4 Outline of the paper, Line 115-121]: The outline is too brief and doesn't provide enough detail about each section's content. Please expand this section to make a clear roadmap. Consider adding a sentence or two about the key points or arguments that will be presented in each section.

Response 7 – We elaborated Section 1.4 with more details of the paper’s roadmap. The new section reads as follows: “The remainder of this paper is organized as follows. Section II summarizes the existing work in the field of gameplay security systems that are behavior-based. Section III details the TVGP protocol, including the protocol’s initiation, setup, connections, and transactions. Section IV presents a security analysis of TVGP, including an analysis of TVGP in temporally both honest/dishonest environments. Section V discusses our experiments for testing sensor detection rate and performance using proof of solution implementation of the TVGP system. Section VI lists the operational domain assumptions of TVGP, including the limitations and potential future work for the TVGP protocol. Finally, Section VII offers our conclusions, followed by acknowledgements and references.”

Comment 8 - [2. Related Work, Line 149]: Update the academic writing for this part: [9], [10], [11], [12], [13], [14], [15], [16], [17], [18], [19], [20], [21], [22], [23], [24], [25], [26], [27], 149 [28]. Using professional writing and citation methods.

Response 8 – We updated the paragraph. The new paragraph reads as follows: “Several literature items have documented networking [9], [10], scalability [11], [12], and load combinatorial issues [13], [14] in a multiplayer or Massively Multiplayer Online (MMO) gaming environments. These works have also tackled cheating issues arising from a technological perspective such as packet sniffing [15], [16], code injection [17], [18], ping fuzzing [19], [20], aim-bots [21], [22], trigger-bots [23], [24], and lag-switching [25], [26]and so on. These works also deal with issues in support mechanisms offered to gamers [27], [28]. Our proposed work deals with behavior-based cheat sensing and can work symbiotically alongside some or many of the works presented in this section”.

Reviewer 2 Report

Comments and Suggestions for Authors

 Referee report

This paper focuses on Trust-Verify Game Protocol (TVGP), which is a sensor protocol intended for low-trust environments and designed to provide game state information to help support behavior-based cheat-sensing detection schemes.  The experimental results validate high efficiency and performance standards for TVGP.  Therefore, I think that the paper makes a contribution and has the potential to be published.  However, I summarize in the GENERAL COMMENTS as follows:

GENERAL COMMENTS

1. On Page 3, there are too many citations in references [9]-[28], and please provide an analysis of these references.

2. TVGP is acknowledged to generate additional system load due to the extra processing and messaging it requires. This could lead to a performance overhead, which might affect the responsiveness of the game, especially if the implementation is not optimized. What optimization strategies are used to avoid introducing latency?

3. The Gameplay section should provide the algorithm flow of the game.

4. In the Security Analysis of Trust-Verify Game Protocol section, the analysis mentions that TVGP can reliably detect most forms of tampering but does not elaborate on the specific mechanisms by which it distinguishes between legitimate and malicious activities. There is a need for a more detailed explanation of how the protocol identifies and differentiates various types of cheating behaviors, especially given the sophisticated nature of modern cheating techniques.

5. In the testing section, the experiments primarily focus on three specific types of behavioral attacks. To thoroughly assess the robustness of TVGP, it would be beneficial to consider a wider variety of attack vectors, such as packet manipulation and other sophisticated exploits that might be encountered in a real-world gaming environment. How to ensure that the system adapts to diversity in attack scenarios?

6. In order to enhance the validity of the testing results, how to expanding the scope of attacks and user participation?

Author Response

We thank the reviewer for their time and valuable feedback. We have addressed the reviewer’s feedback as follows:

Comment 1. On Page 3, there are too many citations in references [9]-[28], and please provide an analysis of these references.

Response 1 - We updated the paragraph. The new paragraph reads as follows: “Several literature items have documented networking [9], [10], scalability [11], [12], and load combinatorial issues [13], [14] in a multiplayer or Massively Multiplayer Online (MMO) gaming environments. These works have also tackled cheating issues arising from a technological perspective such as packet sniffing [15], [16], code injection [17], [18], ping fuzzing [19], [20], aim-bots [21], [22], trigger-bots [23], [24], and lag-switching [25], [26]and so on. These works also deal with issues in support mechanisms offered to gamers [27], [28]. Our proposed work deals with behavior-based cheat sensing and can work symbiotically alongside some or many of the works presented in this section”.

Comment 2. TVGP is acknowledged to generate additional system load due to the extra processing and messaging it requires. This could lead to a performance overhead, which might affect the responsiveness of the game, especially if the implementation is not optimized. What optimization strategies are used to avoid introducing latency?

Response 2 - We updated Section 4.1 to address this feedback. The relevant portions are highlighted in the manuscript.

Comment 3. The Gameplay section should provide the algorithm flow of the game.

Response 3 – The Gameplay section (Section 3.3) is followed by Sections 3.4 and 3.5, which describe the follow of TVGP algorithm. Perhaps the sectioning made it confusing for the reader. We merged Sections 3.4 and 3.5 into 3.3.

Comment 4. In the Security Analysis of Trust-Verify Game Protocol section, the analysis mentions that TVGP can reliably detect most forms of tampering but does not elaborate on the specific mechanisms by which it distinguishes between legitimate and malicious activities. There is a need for a more detailed explanation of how the protocol identifies and differentiates various types of cheating behaviors, especially given the sophisticated nature of modern cheating techniques.

Response 4 - We updated Section 5.1 to address this feedback. The relevant portions are highlighted in the manuscript.

Comment 5. In the testing section, the experiments primarily focus on three specific types of behavioral attacks. To thoroughly assess the robustness of TVGP, it would be beneficial to consider a wider variety of attack vectors, such as packet manipulation and other sophisticated exploits that might be encountered in a real-world gaming environment. How to ensure that the system adapts to diversity in attack scenarios?

Response 5 - We updated Section 5.1 to address this feedback. The relevant portions are highlighted in the manuscript.

Comment 6. In order to enhance the validity of the testing results, how to expand the scope of attacks and user participation?

Response 4 - We updated Section 5.5 to address this feedback. The relevant portions are highlighted in the manuscript.

Round 2

Reviewer 1 Report

Comments and Suggestions for Authors

The authors have adequately addressed all identified comments from the last round review.

The work conveys novel contributions advancing the topic in their research field which will provide value to readers that merits journal dissemination.

Thanks for inviting me as the reviewer.

Reviewer 2 Report

Comments and Suggestions for Authors

Referee report

This paper focuses on Trust-Verify Game Protocol (TVGP), which is a sensor protocol intended for low-trust environments and designed to provide game state information to help support behavior-based cheat-sensing detection schemes.  The experimental results validate high efficiency and performance standards for TVGP.   The authors carefully revised the manuscript and made some changes to the version according to the comments of the reviews. Therefore, I think that the paper makes a contribution and has the potential to be published.
